# Expression Changes of MHC and Other Immune Genes in Frog Skin during Ontogeny

**DOI:** 10.3390/ani10010091

**Published:** 2020-01-06

**Authors:** Quintin Lau, Takeshi Igawa, Shohei Komaki, Yoko Satta

**Affiliations:** 1Department of Evolutionary Studies of Biosystems, Sokendai (The Graduate University for Advanced Studies), Kamiyamaguchi 1560-35, Hayama, Kanagawa 240-0193, Japan; satta@soken.ac.jp; 2Amphibian Research Center, Hiroshima University, 1-3-1, Higashi-Hiroshima, Hiroshima 739-8526, Japan; tigawa@hiroshima-u.ac.jp; 3Division of Biomedical Information Analysis, Iwate Tohoku Medical Megabank Organization, Disaster Reconstruction Center, Iwate Medical University, Idaidori 1-1-1, Yahaba, Iwate 028-3694, Japan; komaki@iwate-med.ac.jp

**Keywords:** anuran development, major histocompatibility complex, *Rana*, *Xenopus*, transcriptome

## Abstract

**Simple Summary:**

Tadpoles undergo many changes in physiology and immunology until metamorphosis into adult frogs. Major histocompatibility complex (MHC) molecules are an important part of vertebrate adaptive immunity, and our study measured the expression of two *MHC* genes (*MHC* class I and II) in skin during six tadpole stages of the Montane Brown frog (*Rana ornativentris*). First, using a qPCR method, we found that both *MHC* class I and II expression significantly increased between stage 24/25 (‘early’) and stage 28 (‘mid’) tadpole skin. Then, we conducted next-generation sequencing for ‘early’, ‘mid’ and ‘late’ stage tadpole skin mRNA of both *R. ornativentris* and a model species, *Xenopus tropicalis*, and confirmed that *MHC* expression increased from the ‘mid’ stage. We also performed further analyses of transcriptome data and found that several immune-related gene ontology terms were upregulated from the ‘mid’ tadpole stage. Our findings probably support that both MHC class I and II have a functional role during tadpole development.

**Abstract:**

Anuran amphibians undergo major physiological and immunological changes following metamorphosis. Genes of the major histocompatibility complex (MHC) code for receptors important for vertebrate adaptive immunity. We used qPCR to measure skin *MHC* expression in six different ontological stages of *Rana ornativentris* (*n* = 10 per stage); normalized *MHC* class I and II expression at the mRNA level was significantly higher in stage 28 (mid-larval) compared to stages 24/25 (early-larval) tadpoles. Subsequent transcriptomic analyses of three tadpole (early-, mid-, and late-larval) stages of *R. ornativentris* and model species *Xenopus tropicalis* focused on mRNA expression of immune-related genes in the skin. Normalized expression of most *MHC* class I and II transcripts in both species were significantly higher in mid- and late-larval stages compared to early-larval stage. In addition, gene ontology (GO) analyses of differentially expressed transcripts revealed several immune-related GO terms that were significantly upregulated from the mid-larval stage. Our study provides evidence that both *MHC* class I and II is expressed during development in both *R. ornativentris* and *X. tropicalis*.

## 1. Introduction

Anuran amphibians have a biphasic life cycle featuring a pre-metamorphic tadpole stage usually inhabiting fully aqueous environments, and a post-metamorphic adult stage inhabiting aqueous and/or terrestrial environments depending on the species. Based on model *Xenopus* frogs, tadpoles are considered to have a poor or immature adaptive immune system [1,2] as well as some components of innate immunity [3,4]. At metamorphosis, anurans are known to go through a complete physiological reorganization; this includes components of the immune system like expression of major histocompatibility complex (MHC) molecules (reviewed by Rollins-Smith, 1998) [5]. The MHC is an essential component of vertebrate adaptive immunity because *MHC* genes code for membrane-bound glycoproteins that recognize, bind and present specific antigens to T lymphocytes. There are two major classes of MHC: MHC class I (MHC-I) molecules predominantly recognize and present endogenous antigenic peptides (e.g., from viruses) to cytotoxic T cells, while MHC class II (MHC-II) molecules present exogenous antigens (e.g., from fungi and bacteria) to helper T cells.

There have been reports of tadpole susceptibility to infectious agents like Perkinsea-like protists [6] and ranaviruses [7], as well as trematode-associated tadpole mortality [8]. Furthermore, a study demonstrated fatality in tadpoles after exposure to chytrid fungus *Batrachochytrium dendrobatidis* (Bd), possibly due to toxins released by the fungus, as well as variation in susceptibility between species [9], although there is no concrete evidence that ties Bd to tadpole mortality. Further research into anuran immune expression and function during ontogeny is vital for broadening our understanding of host-pathogen dynamics, in lieu of the presence of various infectious pathogens that may affect tadpoles.

In general, the expression of MHC in anurans during ontogeny is not widely studied, and has been predominantly restricted to model organism *Xenopus laevis*. A study in *X. laevis* indicated that protein expression of MHC II is generally ubiquitous during ontogeny [10]; specifically, there is early expression in spleen, thymus and B cells, and expression in skin was detected after the mid-larval life stage. It appears that MHC class Ia (classical) expression is low during ontogeny based on northern blot analyses of thymus, spleen and skin mRNA [11,12]. However, a later study of lymphocytes in the spleen and thymus of *X. laevis* mid-stage tadpoles using fluorescence microscopy and flow cytometry supported MHC class I surface expression [13].

Since most studies of tadpole MHC expression have focused on *Xenopus* frogs, applying such findings to all anurans could be premature, considering they are among the most diverse vertebrates with at least 6981 species [14] and with the last common ancestor of Anura appearing around 210 million years ago [15,16]. Adding to this, there is some evidence that *MHC* class I mRNA is expressed in tadpoles of Korean frogs [17]. In our previous transcriptome study of Japanese frogs, we found low normalized expression of both *MHC* class I and II transcripts in a few samples of *Rana japonica* and *R. ornativentris* mid-larval tadpoles (stage 28) [18]. Thus, in this study we wanted to expand on the knowledge related to MHC expression during anuran ontogeny, and conducted an in-depth quantification of mRNA expression levels of *MHC* in a non-model amphibian species. We focused our examination to expression in skin, as it is a barrier between the host and the environment and is also the potential target site entry for some pathogens.

In this study, we quantified mRNA expression levels of *MHC* class Ia and II during various stages of ontogeny of our selected study species, *Rana ornativentris* (Montane Brown frog) using quantitative PCR (qPCR). We subsequently used next-generation sequencing to establish transcriptome data of both *R. ornativentris* and *Xenopus tropicalis* tadpoles, and investigated expression of MHC and differential expression of other immune-related genes.

## 2. Materials and Methods 

### 2.1. Rana ornativentris Samples

A proportion of two separate *Rana ornativentris* egg mass clusters were collected in January 2016 from Yokohama Nature Observation Forest Park, Kanagawa prefecture, Japan (35°20’22” N, 139°35’15” E). Both egg clusters were collected within 72 h after fertilization, and were approved by Sokendai Research committee (approval number 46) and with official permission from the park. The two partial egg clusters (egg cluster A and B) were reared with 12 h:12 h light:dark cycle in separate water containers (42 cm width × 58 cm length × 20 cm depth) containing 15–30 L of aged water with aeration, and water was replaced every 3–7 days. Once tadpoles reached feeding stage (after stage 24), they were fed daily with Pleco Spirulina Wafers (Tetra, Melle, Germany).

Animals (*n* = 5 per stage per egg cluster) were collected at six life/ontological stages (Figure 1, Appendix A) according to those described in the closely related *Rana japonica* [19,20]: (i) stage 20—appearance of external gills, (ii) stage 24/25—external gills covered and appearance of hindlimb buds, (iii) stage 28/29—hindlimb bud is 1.5–2.0 fold of its diameter, (iv) stage 33—appearance of prehallux, (v) stage 37—protrusion of one or both forelimbs and (vi) stage 40—full disappearance of tail. A total of 60 tadpoles were collected for RNA extraction and MHC expression analyses (two egg clusters, six life stages and five animals per stage).

Animals were euthanized by immersion in tricaine methanesulfonate (MS222, 0.5–3 g/L water). Next, skin was immediately dissected for RNA extraction and the remaining body was preserved in RNAlater solution (Applied Biosystems, Carlsbad, CA, USA) at −80 °C. Dissected skin samples were immediately homogenized in ISOGEN (Nippon Gene, Tokyo, Japan) and stored at −80 °C for less than one month before RNA extraction following manufacturer’s protocol. RNA was then synthesized into first-strand complementary DNA (cDNA) using PrimeScript^TM^ RT reagent kit (Takara Bio Inc., Otsu, Japan).

Additional RNA samples, included for investigating MHC expression, were retrospectively extracted from tissues preserved in RNAlater solution using the same isolation procedure and synthesized into cDNA. This included (a) whole bodies of stage 24/25 tadpoles (*n* = 5 animals per cluster) to compare *MHC* expression levels to that of skin in the same stage, and (b) ventral skin (*n* = 6 total) and spleen (*n* = 6 total) of adult frogs collected from Hiroshima prefecture and used in our previous studies [18,21,22].

### 2.2. Primer Design and Generation of Standards for Quantitative PCR (qPCR)

We previously generated transcriptome data from a single *R. ornativentris* adult individual and additionally used molecular cloning in seven individuals to characterize 780 bp and 754 bp sequences for *MHC* class I a (classical) and *MHC* class II beta chain, respectively [18,21]. In the present study, we developed MHC class I and II primers for quantitative polymerase chain reaction (qPCR); we independently aligned all previously characterized *MHC* class I and II variants [18,21], and designed primers in conserved regions that amplified across exon boundaries (Table 1). Primers were designed based on variants from one or more *MHC* class I and class II loci (3–5 *MHC* class I and 1–2 *MHC* class II variants per individual were identified previously), thus our primers may detect all expressed *MHC* class I and II loci in the species. Primers for reference gene glyceraldehyde 3-phosphate dehydrogenase (*GAPDH*, Table 1) were also developed based on conserved regions of assembled contigs from transcriptome data [18] of *R. ornativentris* aligned with that of related species *R. japonica* and *R. tagoi tagoi* (GenBank accession numbers MK541925-MK541927).

To generate standards for each of the three genes, we utilized plasmid DNA generated by traditional PCR and molecular cloning. Traditional PCR amplification was conducted using Applied Biosystems^®^Veriti© thermal cycler in 10 µL reactions containing approximately 100 ng of cDNA (from skin of a stage 40 *R. ornativentris* tadpole), 0.25 U TaKaRa Ex Taq (Takara Bio Inc.), 1X Ex Taq buffer, 0.5 mM each dNTP and 0.5 µM each primer. Cycle condition comprised of 30 cycles of 30-s denaturation at 98 °C, 30-s annealing at 58 °C, and 30-s extension at 72 °C, followed by a final extension of 72 °C for 3 min. Sequences of PCR products were confirmed by sequencing: in brief, they were purified with ExoSAP-IT^®^ (Affymetrix Inc., Santa Clara, CA, USA) and sequenced with BigDye^®^ Terminator Cycle Sequencing Kit (Applied Biosystems, Foster City, CA, USA) and ABI 3130xl automated sequencer. For cloning, PCR products were ligated using T-Vector pMD20 and DNA Ligation Kit 2.0 (Takara Bio Inc.) and incubated for 30 min at 16 °C. Next, ligated reactions were transformed into JM 109 Competent Cells (Takara Bio Inc.) following manufacturer instructions. A single positive clone was cultured overnight in 5 mL of LB broth containing 100 µg/mL ampicillin, and plasmid DNA was purified using QIAprep^®^ Spin Miniprep kit (Qiagen, Hilden, Germany) following manufacturer instructions. From plasmid DNA, we prepared aliquots of standards by 1:10 serial dilutions (10 pg/µL, 1 pg/µL, 0.1 pg/µL, 10 fg/µL and 1 fg/µL).

### 2.3. MHC Expression in R. ornativentris Using Quantitative PCR (qPCR)

We used qPCR to measure expression levels for the two test genes (*MHC* class I and class II) and the reference gene (*GAPDH*) from the 60 *R. ornativentris* tadpole skin samples and additional body and adult samples. Using Thermal Cycler Dice^®^ Real Time System II (Takara), qPCR was conducted in 10 µL triplicate reactions containing 2 µL of template (either standard or sample DNA), 1X SYBR Premix Ex Taq II (Tli RNaseH Plus; Takara Bio Inc.) and 0.8 µM each primer. Reaction conditions were (i) initial denaturation stage: 95 °C for 30-s; (ii) PCR stage: 40 cycles of 95 °C for 5-s and 55 °C for 60-s; (iii) dissociation stage: 95 °C for 15-s then 60 °C for 30-s then 95 °C for 15-s. Quantity was calculated using standard curves and averages from triplicate reactions. Relative expression of either *MHC* class I or II was determined by ratio of quantity divided by that of *GAPDH*.

### 2.4. Statistical Analyses of qPCR Expression Data

All statistical analyses were conducted using R v 3.3.2. All data were log-transformed after quantile–quantile plots and Shapiro–Wilk normality tests confirmed non-normal distribution (data not shown). One-way analysis of variance (ANOVA) was used to compare *MHC* expression between the six tadpole stages, using egg cluster as a block (model: log-*MHC* normalized expression life stage + egg cluster). Subsequently, we used ANOVA to compare *MHC* expression in stage 24/25 tadpole skin with whole body. We also compared *MHC* expression between adult skin and spleen samples to validate our previous expression data extrapolated from transcriptome data, whereby we identified higher normalized expression of *MHC* in spleen relative to skin [18]. We used *p* < 0.05 to denote statistical significance.

### 2.5. Transcriptome Analyses of Immune Expression in R. ornativentris Tadpoles

To validate the changes of *MHC* class I and II expression during ontogeny of *R. ornativentris* tadpoles detected by qPCR and investigate changes in overall immune gene expression, we subsequently conducted transcriptome analyses in a subset of samples: ‘early-larval’ stage 24/25 skin (*n* = 3) and whole body (*n* = 3), ‘mid-larval’ stage 28/29 skin (*n* = 3) and whole body (*n* = 3) and ‘late-larval’ stage 40 skin (*n* = 3). The same RNA samples (2–10 µg of RNA per sample) extracted for qPCR were also used for library construction using NEBNext Ultra Directional RNA Library Prep Kit for Illumina^®^ (New England Bio Labs, MA, USA). Short DNA sequencing (PE150 reads) was conducted at Novogene Co. Ltd. (Beijing, China) using cDNA Illumina HiSeq2000 sequencing system (3 GB per sample). Sequenced reads were filtered by removing reads containing adapters, N > 10%, or low quality (Qscore ≤ 5; Novogene Co. Ltd.). De novo assembly was then conducted with default settings in Trinity version 2.8.3 [24] using combined reads from the *R. ornativentris* libraries to create a reference data set.

Reads from each sample were aligned back to the reference transcript data set using kallisto (v0.45.1) [25] and transcript abundances were reported as transcripts per million (TPM). We isolated MHC class I and II transcripts from the libraries using NCBI-BLAST-2.2.31+ with E-values ≤ 1 × 10^−5^ for blastn search with previously characterized *R. ornativentris* MHC sequences [18,21] as queries, and extrapolated TPM values. We conducted differential gene analyses to test for significant differences in MHC expression and extrapolate other immune-related transcripts with significant expression differences between larval stages (early vs. mid vs. late) or between tissues (skin vs. body). Differentially expressed transcripts (including MHC) between stages or tissues were determined using likelihood ratio test (LRT) implemented in edgeR (3.9) [26]. Pairwise comparisons between tadpole stages or tissues were conducted using false discovery rate (FDR) cut-off of 0.05, and transcripts with log-2-fold-changes logFC > |2| were extracted. All differentially expressed (DE) transcripts were collated and annotated with NCBI-BLAST-2.2.31+ with E-values ≤ 1 × 10^−5^ against *Xenopus tropicalis* 9.1 protein dataset from Xenbase [27] (http://www.xenbase.org/, RRID:SCR_003280). We then submitted the blast results, in the form of GenBank and RefSeq IDs, for gene ontology (GO) enrichment analyses implemented in DAVID 6.8 [28]. All enriched GO terms related to the immune system were collected using a threshold of gene count ≥ 3 and *p* < 0.1.

### 2.6. MHC and Immune Gene Expression in Xenopus Tadpoles

Based on MHC expression data from *Rana ornativentris*, we were interested to explore MHC class expression in *Xenopus* skin mRNA at pre-metamorphic stages. Thus, we also conducted transcriptome analyses of *Xenopus tropicalis* tadpole skin during ontogeny. We collected *X. tropicalis* skin samples from three tadpole stages (*n* = 3 individuals per stage), similar to stages used for *R. ornativentris*. The three stages are stage 40 (‘early-larval’), 50 (‘mid-larval’) and 59 (‘late-larval’) based on *X. laevis* [23], which is equivalent to around stages 21/22, 27 and 37, respectively, in *R. ornativentris*, following inter-species stage comparisons [19,20,29,30] (Appendix A). All individuals were euthanized in a similar approach (see above) and stored in RNAlater solution at −80 °C until RNA extraction and cDNA synthesis, library construction and Illumina^®^ sequencing using the same procedures as in *R. ornativentris*. Using kallisto, clean reads were mapped onto the *X. tropicalis* genome (v9.1 genome assembly), which was downloaded from Xenbase [27]. Differential expression analyses between stages using LRT as well as GO enrichment analyses followed the same procedures as in *R. ornativentris*, while MHC expression levels were extrapolated based on transcripts mapped to *MHC* genes already annotated in the genome.

## 3. Results and Discussion

### 3.1. MHC Expression in R. ornativentris Using qPCR Analyses

Among non-model amphibians, this is one of the first studies to quantify changes in MHC expression during ontogeny. mRNA expression of *MHC* class I in *R. ornativentris* tadpole skin was low during early developmental stages (stage 20, and stage 24/25), and significantly increased in all subsequent stages from stage 28/29 (*p* < 0.00001, Figure 2A). In a similar pattern to *MHC-I*, mRNA expression of *MHC-II* was significantly lower in stages 20 and 24/25 compared to mid- and late-developmental stages (*p* < 0.00001, Figure 2B). Throughout all developmental stages studied, overall expression of *MHC-II* was about 10-fold lower than that of *MHC-I* (Appendix A).

Next, we checked whether the low *MHC* class I and II expression in the skin of stage 24/25 tadpoles was also low in the ‘whole body’ samples of the same individuals. We found that expression was slightly higher in whole body of tadpoles than the skin, but this was statistically significant only for *MHC-II* (*MHC-I*: *p* = 0.0567, *MHC-II*: *p* < 0.001, Figure 2). We also compared expression between spleen and skin from *R. ornativentris* adults (*n* = 6): *MHC-I* expression was not significantly different (*p* = 0.054, Figure 2A) since it is ubiquitously expressed in most mature cells, whereas *MHC-II* was significantly higher in spleen than skin (*p* = 0.022, Figure 2B). These qPCR results were consistent with the normalized expression levels determined from transcriptome data of a single individual in our previous study [18].

### 3.2. MHC Expression in R. ornativentris and X. tropicalis Using Transcriptome Analyses

To validate our findings of higher *MHC* expression in tadpole skin during later stages of ontology from qPCR analyses, we prepared transcriptomic data for three developmental stages of both *R. ornativentris* and *X. tropicalis* (early-, mid- and late-larval stages, with approximately similar stages between the two species, see Appendix A). In *R. ornativentris*, we found two *MHC-I* and one *MHC-II* transcript with higher expression in mid- and late- compared to early-larval stage skin (Table 2); however, this was significant only in the *MHC-II* (M > E: logFC = 3.81) and a single *MHC-I* (M > E: logFC = 2.51; L > E: logFC = 5.38) transcript using a strict corrected cut-off of FDR < 0.05. Our skin-related normalized expression data is comparable to our qPCR results, whereby significantly higher *MHC* class I and II expression in skin was detected at stage 28/29 (mid-larval). However, there were no significant differences between body and skin in *MHC* normalized expression levels based on transcriptome data (Table 2), despite a significant difference in *MHC-II* detected using qPCR (Figure 2B). This may be attributed to differences in normalization and statistical approaches between the two different methods. Nevertheless, there is opportunity in the future to explore MHC expression (gene and protein level) in other specific tissues during tadpole development.

To investigate species differences in *MHC* expression patterns in developmental stages, we also conducted transcriptome analyses in *Xenopus* tadpole skin. In *X. tropicalis*, we found one *MHC-I* and two *MHC-II* transcripts (corresponding to the MHC class II alpha and beta chain) with significantly higher normalized expression in mid- and late- versus early-larval stage skin (log FC = 5.93 − 8.73, FDR < 1E–20). The low *MHC* expression in early-larval and higher expression at mid- and late-larval stage skin was also validated using preliminary standard PCR amplification in both *X. tropicalis* and *X. laevis* (Appendix A).

Our results provide supporting evidence of *MHC* class I expression in *X. laevis* skin mRNA during development using sensitive detection approaches (PCR and especially next-generation sequencing (NGS)). This investigation of *MHC* expression in *Xenopus* tadpoles using NGS approaches provides a springboard for future studies to quantify this expression at a more in-depth level and re-assess expression differences across additional tissues and developmental stages.

### 3.3. Differential Expression of Immune-Related Genes in R. ornativentris and X. tropicalis Tadpoles

In pairwise comparisons of skin samples at different stages (early-, mid- and late-larval stages), we found 740–2008 differentially expressed transcripts in *R. ornativentris* and 79–2708 in *X. tropicalis* (Appendix A). When these transcripts were submitted for GO analyses in DAVID, we found that the wide range of differentially expressed gene numbers from pairwise comparisons were mainly due to the marked differences in cellular, developmental, neural and immunological processes between early- and late-larval stages of both species (Appendix A). When focusing on the ‘biological processes’ parent term, we found 46–402 GO terms in *R. ornativentris* and 28–434 GO terms in *X. tropicalis* that were enriched at specific stages (Appendix A). Among these GO terms, many immune-related terms were enriched in mid- and late-larval stage (24 and 4 terms in *R. ornativentris*, 48 and 38 terms in *X. tropicalis*, respectively) compared to early-larval stage tadpole skin (listed in Figure 3, Appendix A); one term (GO:0019882 antigen processing and presentation), which might be related to MHC function, was enriched in both species. In contrast, only non-immune-related GO terms were enriched in the early-larval stage tadpole skin compared to the latter two stages, including GO:0032501 single-multicellular organism process and GO:0007399 nervous system development (Appendix A).

In *R. ornativentris*, 23 GO terms related to antigen processing and presentation (including GO:0002474 antigen processing and presentation of peptide antigen via MHC class I) were enriched in mid-larval skin compared to early-larval skin (Figure 3A, Appendix A). This adds further support to the significantly increased *MHC* expression, which we detected using both qPCR and transcriptome approaches, especially for class I (Figure 2, Table 2). In *X. tropicalis*, GO:0019882 antigen processing and presentation, GO:0045087 innate immune response, and GO:0002224 toll-like receptor signaling pathway were among the numerous immune-related GO terms significantly enriched in mid- and late-larval skin compared to early-larval skin (Figure 3B, Appendix A). However, no GO terms with specific mention of ‘MHC’ were enriched in this species.

The significant enrichment of immune-related genes from the mid-larval stage, concurrent with significant increases in *MHC* expression, supports that marked development of the tadpole immune system occurs before stage 28 in *R. ornativentris* and stage 50 in *X. tropicalis*. There was only a single immune-related GO term enriched in late-larval versus mid-larval skin of *R. ornativentris* (GO:0002526 acute inflammatory response; Appendix A). Similarly, no immune-related GO terms in skin were significantly enriched after the mid-larval stage of *X. tropicalis*; this suggests that further changes in the immune system do not occur during the remainder of *X. tropicalis* and *R. ornativentris* tadpole development, at least until after metamorphosis. There may also be some other factors that contribute to differences in developmental immunity between the two species: *X. tropicalis* might rely more on the innate rather than adaptive immune system, *R. ornativentris* has a higher number of *MHC* loci (at least three *MHC-I* loci [21]) compared to *Xenopus* (single *MHC-I* locus [31]), and there are species differences in post-metamorphosed life-cycles (semi-aquatic versus fully aquatic). Nevertheless, high-coverage whole genome sequences for *R. ornativentris* would be required for a more comprehensive comparison between species.

Our transcriptomic study focused predominantly on skin samples, although we also examined some ‘whole body’ samples of *R. ornativentris* for a preliminary insight into tissue-specific differences. In comparisons within the same stages of early- and/or mid-larval stage *R. ornativentris* tadpoles, there were more differentially expressed transcripts and enriched GO terms in the body compared to skin (Appendix A). Most of these enriched terms were not immune-related (Appendix A), except GO:0006955 immune response, which was enriched in body compared to skin of combined early- and mid-larval stages (Appendix A). Even so, we have established a foundation and database for further investigation of immune-gene expression in different specific tissues during larval development as well as immune responses to infectious diseases like ranaviruses and Bd.

## 4. Conclusions

We conducted an in-depth analysis of *MHC* class I and II expression in skin of a Japanese frog species during ontogeny using qPCR and next-generation sequencing approaches. We found that expression of *MHC* at the mRNA level was low during early developmental stages, and significantly increased during and following the mid-larval stage. Future immunological assays will be important for validating the functional role of both MHC class I and II during ontogeny in anuran frogs. Analyses of skin transcripts also identified many immune-related GO terms that were upregulated from the mid-larval stage. Our additional examination in *Xenopus* frogs also supports an increased expression of *MHC* and other immune-related genes from the mid-larval stage. Despite the distant evolutionary relationship between *Xenopus* and *Rana* spp., our congruent findings in both support that anuran species have *MHC* expression in the skin during development.

## Figures and Tables

**Figure 1 animals-10-00091-f001:**
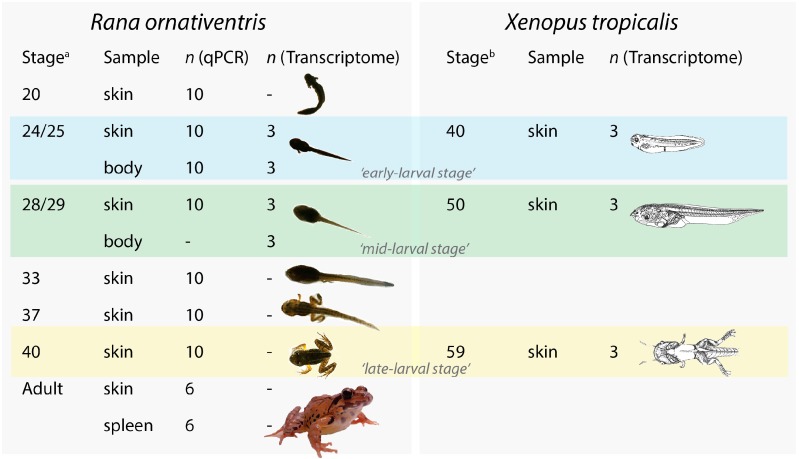
Overview of *Rana ornativentris* and *Xenopus tropicalis* samples used for qPCR and transcriptome analyses in this study. ^a^
*R. ornativentris* stage approximations based on Tahara [19,20]. ^b^
*X. tropicalis* stage approximations based on Nieuwkoop and Faber [23]. Full stage approximations in Appendix A. Images of frogs in all figures are not to scale (source: Q. Lau for *R. ornativentris*, Nieuwkoop and Faber (1994) via Xenbase for *X. tropicalis*).

**Figure 2 animals-10-00091-f002:**
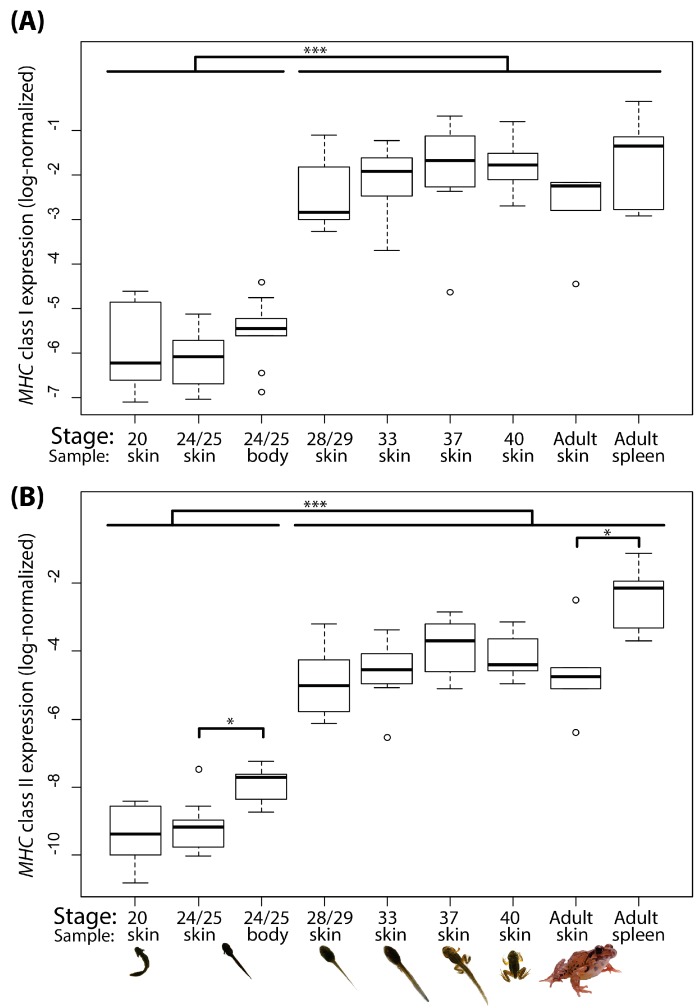
Boxplot of (**A**) *MHC* class I and (**B**) *MHC* class II expression in mRNA of *Rana ornativentris* tadpole skin during six ontological stages and other samples: stage 24/25 body, and adult skin and spleen. *MHC* expression values from qPCR were normalized by the *GAPDH* control gene and log-transformed for normality (log-normalized). * *p* < 0.05, *** *p* < 0.0001.

**Figure 3 animals-10-00091-f003:**
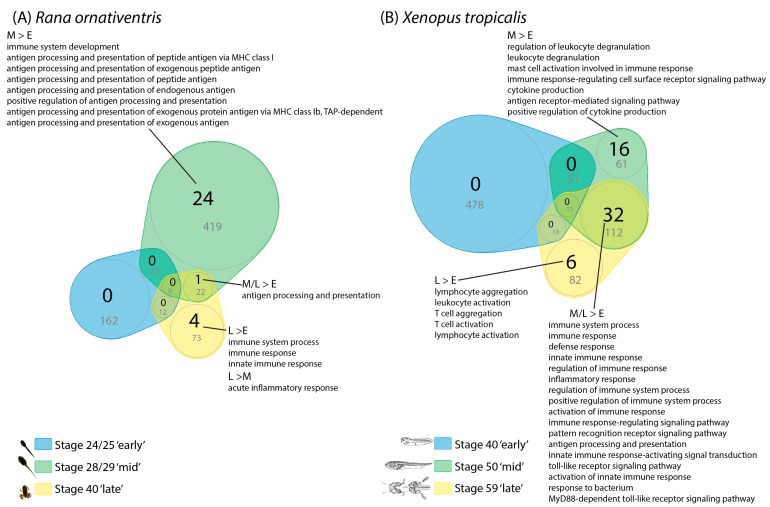
Venn diagrams of gene ontology (GO) terms enriched in skin of early- (E), mid- (M) and late- (L) larval stages from (**A**) *Rana ornativentris* and (**B**) *Xenopus tropicalis*. Number of immune-related enriched GO terms are in black (full list in Appendix A), and total number of enriched GO terms belonging to the ‘biological process’ parent category are in grey and also represented by circle sizes. Examples of enriched immune-related GO terms are listed in comparisons between stages (full list in Appendix A).

**Table 1 animals-10-00091-t001:** Summary of primers used for qPCR in *R. ornativentris*.

Target Gene	Target Species	Boundary Crossed	Forward Primer (5’–3’)	Reverse Primer (5’–3’)	Amplicon Length (bp)
*MHC* class I	*R. ornativentris*	exon 3–4	TCCCGACCATGAATGAGG	GACTGACGATGACCCCACA	190
*MHC* class II beta	*R. ornativentris*	exon 3–4	CACAGCAGCCTGGAGACA	AGCACAAATCCCACAATTCC	98
*GAPDH*	Japanese *Rana*	exon 5–6	CCAACGTGTCTGTGGTTGAC	TCCCAGAATTCCCTTCAGTG	113

**Table 2 animals-10-00091-t002:** Normalized expression of *MHC* class I and II transcripts between different larval stages (E—early, M—mid and L—late, see Appendix A) or tissues of *Rana ornativentris* (Raor) and *Xenopus tropicalis* (Xetr), represented by TPM (transcripts per million) values, and logFC and false discovery rate (FDR) values of pairwise comparisons between samples. Transcripts with significantly different normalized *MHC* expression levels (FDR < 0.05) are indicated in bold.

Sample or Comparison	Raor MHC-I (Transcript 1)	Raor MHC-I (Transcript 2)	Raor MHC-II	Xetr MHC-I [hla-a] (NP_001106536.1)	Xetr MHC-II [hla-dra] (XP_017951884.1)	Xetr MHC-II [hla-drb1] (NP_001039259.1)
*TPM mean ± SD*						
E_body	1.9 ± 1.4	7.7 ± 3.2	0.0 ± 0.0	-	-	-
M_body	18.1 ± 3.4	19.4 ± 8.6	3.1 ± 2.3	-	-	-
E_skin	1.9 ± 0.8	6.7 ± 5.5	0.2 ± 0.2	2.0 ± 2.6	3.0 ± 2.6	43.0 ± 5.6
M_skin	25.3 ± 2.7	50.0 ± 21.2	7.5 ± 4.2	484.3 ± 184.8	741.4 ± 354.3	1957.7 ± 877.9
L_skin	10.8 ± 6.6	41.1 ± 4.9	10.0 ± 7.7	608.5 ± 290.6	711.9 ± 877.9	2374.0 ± 1492.0
*logFC (FDR)*						
skin: E v M	**3.81 (0.008)**	3.00 (0.103)	**2.51 (0.024)**	**8.27 (1.50 × 10^−24^)**	**8.33 (4.58 × 10^−27^)**	**5.93 (4.15 × 10^−21^)**
skin: E v L	2.37 (0.400)	2.55 (0.297)	**5.38 (0.037)**	**8.73 (5.57 × 10^−47^)**	**8.33 (3.92 × 10^−43^)**	**6.30 (1.87 × 10^−40^)**
skin: M v L	−1.40 (0.550)	−0.46 (1)	0.27 (1)	0.49 (1)	0.04 (1)	0.39 (1)
body v skin: E	−0.02 (1)	−0.29 (1)	4.70 (1)	-	-	-
body v skin: M	0.63 (0.904)	1.54 (0.465)	1.39 (0.682)	-	-	-
body v skin: E/M	0.58 (1)	1.16 (0.751)	1.46 (0.922)	-	-	-

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
