# Peer review of "Expression Changes of MHC and Other Immune Genes in Frog Skin during Ontogeny"

_animals, 2020, doi:10.3390/ani10010091_

Round 1

Reviewer 1 Report

Overview: For the most part, this manuscript appears to be technically sound and adds some new information about the possible expression of major histocompatibility (MHC) antigens in a Japanese Ranid frog (Rana ornativentris).  My only caution is that the authors appear to suggest that their findings are quite novel and seem to challenge accepted dogma concerning the expression and function of MHC class I antigens in developing tadpoles.  The authors have overlooked previous studies that show more clearly by flow cytometry that MHC class I antigens are indeed expressed in some lymphocytes of Xenopus laevis tadpoles early in development. The problem is that expression is very low until near the time of metamorphosis.  This suggests that even if present, the function of MHC class I antigens in shaping the lymphocyte repertoire is uncertain.  Specific concerns are listed below.

Specific Criticisms:

Lines 36-37.  You suggest that your findings by qPCR of message for MHC class I and II suggests functional roles that seem to contradict past studies of low MHC class I expression in   You have not really shown functional roles for MHC class I in R. ornativentris.  You have not even shown by flow cytometry that complete molecules are expressed on the surface of the cells.  This would be stronger support for the hypothesis that MHC class I on antigen presenting cells have an important role to play. My point is that you have suggestive evidence that MHC is expressed in the tadpoles of R. ornativentris, and this is of interest.  But it is not surprising and does not really challenge previous findings in Xenopus.  See for example, Rollins-Smith et al. 1997. Developmental Immunology 5:133-144.  Even at very young larval stages, some lymphocytes express MHC class I. Lines 244-246. I’m not sure of the reason to look for statistically different expression of class I in spleen versus skin of adults.  MHC class I in adults would be expected to be present on all adult tissues at comparable levels.  If you compared on a cell per cell basis, it should be about the same. Line 267. I think you mean to use the word “comparable” rather than “comparative”.  You want to say they are similar or about the same. Lines 339-346. You seem to overstate the novelty of your findings.  There is a great deal of evidence in Xenopus as well as other species that immune genes must be expressed because there are lymphocyte-dependent functions that are lost or reduced when the T lymphocyte compartment (by thymectomy) is ablated.  I think it is premature to suggest based on qPCR that basal MHC expression in late larval life is about the same as in adults. Because most people will not have immediate access to the supplementary tables, it would be important to list some of the important immune process genes identified in the transcriptomic studies within the main body of the manuscript. Just showing total number is not very useful. Do we know the identity of any of these genes?

Author Response

Thank you Reviewer 1 for your constructive feedback. We apologize that we overlooked the previous study supporting MHC class I expression through flow cytometry. We have now removed and reduced emphasis on all sentences referring to lack of MHC I expression and importance during frog development. Please see details below. All line references refer to the revised manuscript titled ‘animals-672474_20191231.docx ‘ which contains tracked changes. All reviewer comments are presented in blue italic text.

Specific Criticisms:

Lines 36-37.  You suggest that your findings by qPCR of message for MHC class I and II suggests functional roles that seem to contradict past studies of low MHC class I expression in   You have not really shown functional roles for MHC class I in R. ornativentris.  You have not even shown by flow cytometry that complete molecules are expressed on the surface of the cells.  This would be stronger support for the hypothesis that MHC class I on antigen presenting cells have an important role to play. My point is that you have suggestive evidence that MHC is expressed in the tadpoles of R. ornativentris, and this is of interest.  But it is not surprising and does not really challenge previous findings in Xenopus.  See for example, Rollins-Smith et al. 1997. Developmental Immunology 5:133-144.  Even at very young larval stages, some lymphocytes express MHC class I.

We apologize for this oversight and have markedly down-played the focus of this point (i.e. MHC I expression not detected /important during tadpole development), and incorporated the important reference that you have suggested, e.g.:

-Line 82-88: “It appears that MHC class Ia (classical) expression is low during ontogeny [11,12]: using  based on northern blot analyses of, no mRNA expression was found in  thymus, spleen and skin mRNA [11,12]; and expression levels in these tissues increased only after the metamorphic climax [12]. However, a later study of lymphocytes in the spleen and thymus of X. laevis mid-stage tadpoles using fluorescence microscopy and flow cytometry supported MHC class I surface expression [13].”

-Deleted sentences in line 313-317

Lines 244-246.

I’m not sure of the reason to look for statistically different expression of class I in spleen versus skin of adults.  MHC class I in adults would be expected to be present on all adult tissues at comparable levels.  If you compared on a cell per cell basis, it should be about the same.

Thank you for raising up this point. Yes, there is no major need to investigate tissue-expression differences since MHC class I is expressed in most adult/mature cells. Although comparing MHC-II expression is still informative. Based on your comment, we have rewritten these lines:

“We also compared expression between spleen and skin from R. ornativentris adults (n = 6): MHC-I expression was not significantly different (p = 0.054, Figure 2A) since it is ubiquitously expressed in most mature cells, whereas MHC-II was significantly higher in spleen than skin (p = 0.022, Figure 2B).” (Lines 252-265)

Line 267. I think you mean to use the word “comparable” rather than “comparative”.  You want to say they are similar or about the same.

Yes, you are right, we have changed this word (line 284)

Lines 339-346. You seem to overstate the novelty of your findings.  There is a great deal of evidence in Xenopus as well as other species that immune genes must be expressed because there are lymphocyte-dependent functions that are lost or reduced when the T lymphocyte compartment (by thymectomy) is ablated.  I think it is premature to suggest based on qPCR that basal MHC expression in late larval life is about the same as in adults.

This is a valid comment from Reviewer 1, and we have decided to delete this paragraph (see line 360-367).

Because most people will not have immediate access to the supplementary tables, it would be important to list some of the important immune process genes identified in the transcriptomic studies within the main body of the manuscript. Just showing total number is not very useful. Do we know the identity of any of these genes?

In the original manuscript, some of the immune process genes were already listed in Figure 2 of the original submission. Now, we have markedly expanded this list of immune-related GO terms in the same figure (now Figure 3 in revised manuscript).

Reviewer 2 Report

Major comments

     Expression of MHC and other immune genes is remarkably important in frogs against chytridiomycosis. The authors investigated expression and changes of the genes in the Montane Brown frog skin during ontogeny by qPCR and next-generation sequencing. They further conducted transcriptome and GO analyses, comparing with the model frog, Xenopus tropicalis, and clarified that the gene-expression level has increased from mid-larval stages. These findings are enough worth to be published at the first step, whereas function of the genes is still unknown in the non-model frog.

Minor comments

     The reviewer recommends the following expression.

Line 16     tadpoles > tadpole

Line 23     seem to support > probably support

Abstract

       Some prepositions, for example Line 25 “during”, Line 26 “from, until”, Line 28 “from”, Line 29 “between”) are unsuitable and the sentences are somehow difficult to understand. The authors should revise the sentences carefully.

Figure 2 The letter sizes of GO terms are too small to see. Larger sizes of letters should be used.

Line370      and > mistype? should be deleted

Author Response

Thank you for taking the time to read and review our manuscript, and for the positive support. All line references refer to the revised manuscript titled ‘animals-672474_20191231.docx ‘ which contains tracked changes. All reviewer comments are presented in blue italic text.

Line 16     tadpoles > tadpole

Changed (line 16)

Line 23     seem to support > probably support

Changed (line 23)

Abstract: Some prepositions, for example Line 25 “during”, Line 26 “from, until”, Line 28 “from”, Line 29 “between”) are unsuitable and the sentences are somehow difficult to understand. The authors should revise the sentences carefully.

We have revised these sentences to improve clarity and readability (lines 25-26, and 28-31).

Figure 2 The letter sizes of GO terms are too small to see. Larger sizes of letters should be used.

We have increased the letter size of the GO terms in the now Figure 3. In addition, based on feedback from Reviewer 1, we have markedly increased this list.

Line370      and > mistype? should be deleted

Sorry, we are not sure what this minor comment is referring to.

Reviewer 3 Report

Title: Expression changes of MHC and other immune genes in frog skin during ontogeny

Authors Lau et al.

Lau et al present work that investigates MHC class I, MHC class II, and general gene expression in tissues of anuran tadpoles throughout development. The authors note that MHC class I and II expression begins at the transition between early-larval and mid-larval stages. This pattern was confirmed by RNAseq with MHC and other immune related genes. This study provides novel information into the development of anuran immune systems throughout metamorphosis.

I think that this paper is very well written. I have no issues with the methodology used to derive gene expression data. The study design is great, and I believe that the results will be important to both herpetologists and developmental biologists. However, I must disagree with the way the authors have anchored the paper. The authors justify their use of skin tissue for gene expression of MHC class I and class II based on the threat of the pathogen Batrachochytrium dendrobatidis (Bd). While chytridiomycosis (the disease caused by Bd) is negatively affecting amphibians throughout the world, I do not think that it should be the hook of this paper. As the authors state in line 66, evidence shows that Bd infects keratinized tissue. In tadpoles, Bd infections are localized to the only keratinized tissue in the body (mouthparts). The paper cited by the authors to justify negative effects on tadpoles by Bddoes not support the idea that Bd can negatively affect tadpoles (yes, it provides some examples but there is no concrete evidence that Bd is directly tied to tadpole mortalities). Thus, I doubt that MHC II molecules in the skin of tadpoles would come in contact with Bd. As such, I think that the authors should downplay Bd, and focus on pathogen threats that are lethal to tadpoles (e.g., Ranavirus, trematodes, Perkinsea).

Specific comments are outlined below:

Line 14: I suggest the authors change the word ‘receptors’ to ‘proteins’ or ‘molecules’

Line 17: add ‘expression’ between ..”and II” and “significantly increased.”

Methods section: This section might benefit from a figure that displays all the stages used in your study and what tissues/analyses where collected/performed at each stage.

Author Response

Thank you Reviewer 3 for taking the time and effort to review our manuscript and providing constructive critical feedback. Below is a point-by-point response to all comments. All line references refer to the revised manuscript titled ‘animals-672474_20191231.docx ‘ which contains tracked changes. All reviewer comments are presented in blue italic text.

Throughout the revised manuscript, we have downplayed the emphasis on Bd. For example, we have markedly reduced the paragraph in introduction (starting in line 59) to:

“There have been reports of tadpole susceptibility to infectious agents like Perkinsea-like protists [6] and ranaviruses [7], as well as trematode-associated tadpole mortality [8]. Furthermore, a study demonstrated fatality in tadpoles after exposure to chytrid fungus Batrachocytridium dendrobatidis (Bd), possibly due to toxins released by the fungus, as well as variation in susceptibility between species [9], although there is no concrete evidence that ties Bd to tadpole mortality. Further research into anuran immune expression and function during ontogeny is vital for broadening our understanding of host-pathogen dynamics, in lieu of the presence of various infectious pathogens that may affect tadpoles.”

Specific comments:

Line 14: I suggest the authors change the word ‘receptors’ to ‘proteins’ or ‘molecules’

Changed, see line 14.

Line 17: add ‘expression’ between ..”and II” and “significantly increased.”

Changed, see line 17.

Methods section: This section might benefit from a figure that displays all the stages used in your study and what tissues/analyses where collected/performed at each stage.

Based on your suggestion, I have added a new figure that summarizes all samples and stages used in this study. (See new Figure 1)